# Prospects and Challenges of Lung Cancer Vaccines

**DOI:** 10.3390/vaccines13080836

**Published:** 2025-08-05

**Authors:** Zhen Lin, Zegang Chen, Lijiao Pei, Yueyun Chen, Zhenyu Ding

**Affiliations:** 1Department of Biotherapy, Cancer Center and State Key Laboratory of Biotherapy, West China Hospital, Sichuan University, Chengdu 610041, China; linzhenzlk@wchscu.edu.cn (Z.L.); peilijiao1701@wchscu.edu.cn (L.P.); chenyueyun@wchscu.edu.cn (Y.C.); 2Department of Oncology, Chongqing Liangping District People’s Hospital, Liangping, Chongqing 405200, China; chenzegangd@163.com

**Keywords:** cancer vaccines, immunotherapy, lung cancer

## Abstract

Lung cancer remains one of the most prevalent and lethal malignancies worldwide. Although conventional treatments such as surgery, chemotherapy, and radiotherapy have modestly improved patient survival, their overall efficacy remains limited, and the prognosis is generally poor. In recent years, immunotherapy, particularly immune checkpoint inhibitors, has revolutionized cancer treatment. Nevertheless, the immunosuppressive tumor microenvironment, tumor heterogeneity, and immune escape mechanisms significantly restrict the clinical benefit, which falls short of expectations. Within this context, cancer vaccines have emerged as a promising immunotherapeutic strategy. By activating the host immune system to eliminate tumor cells, cancer vaccines offer high specificity, low toxicity, and the potential to induce long-lasting immune memory. These advantages have positioned them as a focal point in cancer immunotherapy research. This paper provides a comprehensive overview of recent clinical advances in lung cancer vaccines, discusses the major challenges impeding their clinical application, and explores potential strategies to overcome these barriers.

## 1. Introduction

Lung cancer is a leading cause of cancer-related morbidity and mortality worldwide. In 2022, approximately 2.48 million new cases and 1.81 million deaths were reported globally [1], posing a substantial threat to public health and placing a heavy burden on healthcare systems. Currently, surgery, chemotherapy, and radiotherapy remain the primary treatment modalities. Although targeted therapies have significantly improved the prognosis of patients with driver gene mutations [2,3,4], the five-year survival rate for patients with acquired resistance [5] or without gene mutations [6] remains only 10–19% [7,8]. Immunotherapy has emerged as a transformative approach in cancer treatment, encompassing immune checkpoint inhibitors (ICIs), adoptive cell therapy (ACT), and cancer vaccines. Among these, ICIs have demonstrated favorable efficacy in a subset of lung cancer patients [9,10,11,12]. Nonetheless, due to tumor heterogeneity, an immunosuppressive microenvironment, and evasion mechanisms, ICIs face considerable challenges, including high rates of resistance and limited objective response, which severely constrain their clinical effectiveness. Therefore, there is an urgent need to optimize existing immunotherapeutic strategies [13].

Cancer vaccines exert antitumor effects by activating the immune system to recognize and attack cancer cells [14]. The process mainly involves antigen recognition and processing, T-cell activation, and immune-mediated cytotoxicity. Tumor antigens in the vaccine are captured and processed by antigen-presenting cells (APCs), which present them via major histocompatibility complex (MHC) molecules to T cells. The activated APCs migrate to lymph nodes, where they interact with CD8^+^ and CD4^+^ T cells, inducing cytotoxic T lymphocytes (CTLs) to secrete perforin and other effector molecules that directly kill tumor cells. CD4^+^ T cells also release cytokines to enhance CTL function. Moreover, cancer vaccines promote the generation of memory T cells, offering durable immune protection and reducing the risk of tumor recurrence (Figure 1). Compared to conventional therapies, cancer vaccines offer several advantages, including high antigen specificity, low systemic toxicity, and the capacity to induce long-term immune memory, making them a compelling direction for cancer immunotherapy development [15].

The history of tumor vaccines can be traced back to the late 19th century, when William B. Coley used a mixture of bacterial toxins, later known as Coley’s toxins, to treat advanced cancers [16]. However, research and development in this field progressed slowly over the subsequent decades. Significant breakthroughs in tumor vaccine development did not begin to emerge until the late 20th century, propelled by rapid advances in immunology and molecular biology, as well as extensive research into tumor-associated antigens (TAAs) and tumor-specific antigens (TSAs) [17]. To date, the U.S. Food and Drug Administration (FDA) has approved several cancer vaccines for prevention or treatment, including Bacillus Calmette–Guérin (BCG) for bladder cancer (1990) [18], the human papillomavirus (HPV) vaccine for cervical cancer (2006) [19], Sipuleucel-T for prostate cancer (2010) [20], and T-VEC for melanoma (2015) [21].

However, the development of lung cancer vaccines has lagged behind that of vaccines for other solid tumors. Several large phase III trials, such as START [22], MAGRIT [23], and the Belagenpumatucel-L study [24], failed to achieve their primary endpoints, casting doubt on the clinical viability of lung cancer vaccines. These failures were primarily due to suboptimal antigen selection, poor immunogenicity, ineffective delivery systems, and inadequate patient stratification strategies, underscoring the limitations of traditional vaccine design in the context of the highly heterogeneous lung cancer landscape.

Recent advances in high-throughput sequencing technologies, such as next-generation sequencing (NGS), and improved neoantigen prediction algorithms [25] have facilitated the shift toward precision and personalized vaccine development. Personalized neoantigen vaccines have shown promising clinical efficacy in multiple solid tumors [26,27,28]. Concurrently, innovations in vaccine delivery platforms—including lipid nanoparticles (LNPs) [29], viral vectors [30], polymer-based nanoparticles, and self-assembling nanomaterials [31]—have enhanced the stability, targeting, and immunogenicity of antigenic or nucleic acid-based components in vivo. These delivery technologies have not only improved antigen expression and presentation but also provided essential technical support for the rapid manufacturing and clinical translation of personalized vaccines [32]. Furthermore, combining lung cancer vaccines with ICIs, radiotherapy, or targeted therapies has shown synergistic potential and may further improve clinical outcomes. This review aims to provide a comprehensive summary of recent clinical progress in lung cancer vaccine development, highlight current challenges, and discuss future perspectives in this promising field.

## 2. Types of Lung Cancer Vaccines and Advances in Clinical Research

Tumor vaccines can be broadly classified into two categories based on their intended purpose: preventive vaccines and therapeutic vaccines. In the field of lung cancer, current vaccine development is primarily focused on therapeutic vaccines as a potential enhancement strategy in immunotherapy. Furthermore, based on formulation and biochemical characteristics, lung cancer vaccines can be subdivided into nucleic acid vaccines, peptide vaccines, cell-based vaccines, and viral vector vaccines (Table 1). The following sections provide a systematic overview of each category.

## 3. Nucleic Acid Vaccines

Nucleic acid vaccines work by introducing genetic material encoding tumor antigens into the host to elicit an immune response capable of eliminating tumor cells. These vaccines primarily include messenger ribonucleic acid (mRNA) and deoxyribonucleic acid (DNA) vaccines. They offer several advantages, such as flexible design, rapid development timelines, and relatively low manufacturing costs [33].

### 3.1. mRNA Vaccines

The history of mRNA vaccines dates back to 1990, when Wolff et al. first demonstrated that intramuscular injection of naked mRNA encoding β-galactosidase in mice led to successful cellular uptake, in vivo translation into protein, and the induction of an immune response, marking the first demonstration of mRNA as a vaccine vector [34]. Five years later, Conry RM and colleagues extended this approach to cancer immunotherapy by delivering mRNA encoding carcinoembryonic antigen (CEA), successfully inducing antigen-specific antibodies and cytotoxic T lymphocytes (CTLs) in murine models, thus validating the feasibility of mRNA-based cancer vaccines [35]. However, early development efforts were hindered by challenges such as mRNA instability, limited in vivo delivery efficiency, and insufficient production technologies [36].

In the 21st century, advances in molecular biology and delivery platforms significantly propelled the development of mRNA vaccines. Enhancements such as 5′ capping, extended 3′ polyadenylation, and modified nucleotides (e.g., substitution of uridine with N1-methyl-pseudouridine) greatly improved mRNA stability and translation efficiency while minimizing innate immunogenicity [37]. In parallel, the development of LNP delivery systems effectively addressed the challenge of in vivo delivery, improving both targeting precision and cellular uptake [38]. The Coronavirus Disease 2019 (COVID-19) pandemic further accelerated technological maturity and large-scale production capabilities of mRNA vaccines, particularly the optimization of LNP formulations, thereby catalyzing a new wave of cancer vaccine development [39,40].

Among the leading mRNA-based candidates is V940 (mRNA-4157), a personalized neoantigen vaccine developed by Moderna. In exploratory studies such as the KEYNOTE-603 trial (NCT03313778), V940, administered alone or in combination with pembrolizumab, exhibited favorable safety, tolerability, and immunogenicity in patients with advanced solid tumors, including non-small cell lung cancer (NSCLC), and induced neoantigen-specific T cell responses [41]. Subsequently, the KEYNOTE-942 trial (NCT03897881), a Phase IIb randomized controlled study, evaluated V940 plus pembrolizumab versus pembrolizumab monotherapy as adjuvant therapy for resected high-risk stage IIIB–IV melanoma. The combination significantly improved recurrence-free survival (RFS), reducing the risk of recurrence or death (HR = 0.561; 95% CI: 0.309–1.017; *p* = 0.0266), and also showed a trend toward improved distant metastasis-free survival (DMFS) [42]. Encouraged by these results, a Phase III randomized controlled trial (INTerpath-002, NCT06077760) is now underway to assess V940 combined with pembrolizumab versus pembrolizumab alone as adjuvant therapy for NSCLC. The trial design was presented at the 2024 ASCO Annual Meeting, and further data are awaited [43].

Another promising candidate is CV9201, developed by CureVac, which encodes five tumor-associated antigens (TAAs) relevant to NSCLC. In a Phase I/IIa clinical trial involving patients with advanced NSCLC, CV9201 was well tolerated and elicited immune responses against at least one antigen in 63% of participants. Although no objective tumor responses were observed, the median progression-free survival (PFS) was 5.0 months and the median overall survival (OS) was 10.8 months, suggesting potential clinical benefit and a need for further optimization to enhance immunogenicity and efficacy [44]. Building on this, CV9202 (BI 1361849), encoding additional TAAs (e.g., NY-ESO-1, MAGE-C1), was developed and evaluated in a Phase Ib trial in combination with localized radiotherapy in patients with advanced NSCLC and stable disease. This study demonstrated good tolerability and enhanced immune activation [45]. An ongoing clinical trial (NCT03164772) is currently assessing CV9202 in combination with durvalumab, with or without tremelimumab, in advanced NSCLC; results are pending publication. These studies underscore the potential of mRNA vaccines to act as immunomodulators that activate immune responses and reprogram the tumor microenvironment (TME), thereby converting “cold” tumors into “hot” tumors to enhance antitumor efficacy [46].

BNT116 is an off-the-shelf mRNA vaccine developed by BioNTech using its FixVac platform. It encodes six shared TAAs commonly expressed in NSCLC (MAGE-A3, CLDN6, KK-LC-1, PRAME, MAGE-A4, and MAGE-C1), with the aim of eliciting broad-spectrum immune responses. BNT116 is currently being evaluated in several trials, including LuCa-MERIT-1 (NCT05142189), a Phase I/II trial investigating its use in combination with cemiplimab or chemotherapy as first-line treatment for advanced NSCLC, and NCT05557591, a Phase I dose-escalation study involving patients with advanced or metastatic solid tumors, including NSCLC.

In addition to these representative candidates, several ongoing trials are exploring personalized neoantigen mRNA vaccines tailored to the individual tumor mutational landscape in NSCLC. These include NCT03908671 (a Phase II trial evaluating autologous mRNA dendritic cell vaccines plus PD-1 inhibitors), NCT06735508, and NCT06685653. The goal of these studies is to achieve precise and personalized immune activation by targeting patient-specific tumor antigens.

Despite the promising potential of mRNA vaccines in lung cancer therapy, several key challenges remain. These include the need to improve tumor antigen selection and validation, enhance mRNA stability and delivery efficiency, and overcome the complex immunosuppressive mechanisms within the TME. Future research should focus on optimizing vaccine design, exploring synergistic combinations with ICIs, radiotherapy, or targeted therapies, and developing more effective immune modulation strategies to fully realize the clinical potential of mRNA vaccines and improve patient outcomes.

### 3.2. DNA Vaccines

DNA vaccines function by delivering DNA encoding tumor antigens into the host, where the antigens are expressed by host cells, processed by APCs, and presented to T cells, thereby eliciting a specific immune response against tumor cells. Compared to the increasingly prominent mRNA vaccines, DNA vaccines offer notable advantages attributed to their double-stranded structure, including high chemical stability, simplified manufacturing and purification processes, reduced production costs, and the ability to be stored and transported at ambient temperatures, factors that collectively enhance their global accessibility [47]. Leveraging these properties, DNA vaccines have been investigated in various solid tumors such as melanoma [48], breast cancer [49], prostate cancer [50], and cervical cancer [51], providing both theoretical and empirical support for their application in lung cancer.

As a promising immunotherapeutic strategy, DNA vaccines are gaining traction in lung cancer research, with emerging preclinical and early clinical evidence supporting their safety and efficacy. For example, Weng et al. [52] developed a DNA vaccine targeting kirsten rat sarcoma viral oncogene homolog (KRAS) driver mutations commonly found in lung cancer, which successfully induced antitumor immune responses in a murine model harboring this mutation, thereby validating the feasibility of targeting key oncogenes. Zhang et al. [53] designed a vaccine that targets type I conventional dendritic cells, capitalizing on their efficient cross-presentation capacity to activate CD8^+^ T cells. In mouse models of lung cancer, the vaccine demonstrated synergistic efficacy when combined with gemcitabine and ICIs, underscoring the importance of optimized antigen delivery and combination regimens.

These advances have accelerated the clinical translation of DNA vaccines in lung cancer. Several clinical trials are currently underway to assess their efficacy in patients with various histologic subtypes and disease stages. For instance, NCT04397003 is a phase II trial in patients with extensive-stage small cell lung cancer (ES-SCLC), evaluating a personalized neoantigen DNA vaccine—designed based on individual mutational profiles—in combination with durvalumab as maintenance therapy following induction treatment with standard chemotherapy plus durvalumab. This approach aims to induce tumor-specific T cell responses, overcome the immunosuppressive TME, and prolong therapeutic benefit. Another trial, NCT05242965, targets advanced NSCLC and evaluates the multiepitope DNA vaccine STEMVAC. This vaccine includes antigens such as CD105, YB-1, SOX2, CDH3, and MDM2, which are associated with tumor stemness, metastasis, and treatment resistance, with the goal of eliciting a broad-spectrum immune response. STEMVAC is administered as maintenance therapy after standard chemo-immunotherapy, and its immunogenicity is assessed by tracking dynamic changes in peripheral CD8^+^ tumor-infiltrating lymphocytes (TILs), with comparisons made between administration alone and in combination with the adjuvant granulocyte-macrophage colony-stimulating factor (GM-CSF).

Despite their promise, DNA vaccines face substantial challenges in clinical translation. The most critical limitation is insufficient immunogenicity, often resulting from low efficiency in plasmid uptake across cellular and nuclear membranes, rapid degradation by cytoplasmic nucleases, and a lack of robust immune-activating signals [54,55]. Additionally, poor antigen expression in APCs, combined with the immunosuppressive TME characteristic of lung cancer—marked by regulatory T cells (Tregs), myeloid-derived suppressor cells (MDSCs), and overexpression of immune checkpoint molecules—may further attenuate vaccine-induced antitumor responses [56]. To improve the therapeutic efficacy of DNA vaccines, future research should focus on three key areas:

I. Antigen design and optimization: Integrating bioinformatics, genomics, and proteomics to identify lung cancer-specific TSAs and personalized neoantigens with high immunogenicity. Engineering strategies such as multiepitope fusion and signal peptide optimization can enhance antigen processing and presentation, broaden the immune response repertoire, and reduce the risk of immune escape [57].

II. Delivery system innovation: Improving plasmid design (e.g., through stronger promoters or inclusion of CpG motifs to activate innate immunity) [58,59] and developing advanced delivery platforms such as LNPs, virus-like particles (VLPs), or engineered bacteria. These can be further enhanced with physical methods like in vivo electroporation to improve DNA uptake and targeting efficiency [60].

III. Adjuvant strategy refinement: Combining DNA vaccines with potent immunostimulatory adjuvants—such as STING agonists—to enhance dendritic cell activation, facilitate cytotoxic T cell responses, and promote memory T cell formation, ultimately driving a more robust and durable adaptive immune response [61].

## 4. Peptide Vaccines

Peptide vaccines are designed to elicit tumor-specific immune responses by introducing short synthetic peptides derived from TAAs, TSAs or neoantigens. Due to their ease of synthesis, high safety profile, good stability, and adaptability to personalized design, peptide vaccines hold great promise in cancer immunotherapy [62].

### 4.1. Peptide Vaccines Targeting Specific Antigens

Targeting a single, well-defined antigen is the classical approach in peptide vaccine development. In ALK-positive NSCLC, treatment typically relies on tyrosine kinase inhibitors (TKIs) [63], while ICIs have shown limited efficacy, and resistance mechanisms remain incompletely understood [64]. A preclinical study demonstrated that an ALK-specific peptide vaccine could induce robust antitumor immune responses and significantly enhance pulmonary tumor clearance when combined with TKIs, offering a novel immunotherapeutic strategy for ALK-positive patients [65]. Another promising target is human telomerase reverse transcriptase (hTERT). In a phase I clinical trial, the hTERT-derived peptide vaccine UV1 demonstrated favorable safety and immunogenicity in patients with advanced NSCLC, with 67% of participants developing specific T-cell responses. Notably, these responses correlated in a dose-dependent manner with overall survival [66]. CIMAvax-EGF, a vaccine targeting epidermal growth factor (EGF), significantly prolonged median OS (12.43 vs. 9.43 months, *p* = 0.036) as maintenance therapy in a phase III trial for NSCLC [67]. Indoleamine 2,3-dioxygenase (IDO), a key enzyme involved in regulating the tumor immune microenvironment and promoting immune escape [68], has also been explored as a target. A phase I trial evaluating IDO-targeted peptide therapy in advanced NSCLC demonstrated long-term disease control in some patients with a favorable safety profile [69].

### 4.2. Universal or Multi-Antigen Peptide Vaccines

Peptide vaccines incorporating multiple epitopes have also been investigated in NSCLC. In a phase I/II clinical study, researchers identified a set of shared tumor antigens in NSCLC patients resistant to immunotherapy and developed a long peptide vaccine, TEIPP24, based on lipoprotein receptor-related protein associated protein 1 (LRPAP1). The vaccine showed good safety and elicited immune responses in 26 participants [70]. OSE2101 is a multi-peptide vaccine targeting antigens such as HER2, CEA, MAGE2/3, and p53. In immunotherapy-resistant NSCLC patients, it significantly improved overall survival (OS) in the secondary resistance subgroup (11.1 vs. 7.5 months, *p* = 0.017) and also enhanced quality of life [71]. A phase III randomized controlled trial (NCT06472245) is ongoing to further evaluate its efficacy in HLA-A2-positive NSCLC patients with immunotherapy resistance. In another phase I trial, a multi-peptide vaccine composed of four HLA-A24-restricted peptides (derived from VEGFR1, VEGFR2, URLC10, and TTK/CDCA1) induced T-cell responses in 87% of participants, with some achieving disease stabilization [72]. UCPVax, a vaccine targeting a universal tumor antigen, demonstrated good immunogenicity and a 39% disease control rate in a phase Ib/IIa study. Responders experienced significantly better survival compared to non-responders [73].

### 4.3. Personalized Neoantigen Peptide Vaccines

NEO-PV-01 is a personalized neoantigen vaccine. In a phase I study, it was shown to induce durable T-cell responses and promote tumor infiltration when combined with nivolumab in patients with advanced NSCLC [74]. Further studies indicated that when used as a first-line therapy in combination with chemotherapy and ICIs for non-squamous NSCLC, NEO-PV-01 could induce CD4^+^ T cells with effector and cytotoxic features, highlighting its potential in reshaping the tumor immune microenvironment [75].

Given their high safety, design flexibility, and strong antigen specificity, peptide vaccines have become a key area of research in NSCLC immunotherapy. Notable progress has been made across three major strategies: targeting specific antigens, multi-antigen or shared antigen vaccines, and personalized neoantigen vaccines. Several candidates have demonstrated promising immunogenicity and preliminary clinical efficacy in early-phase trials, with some patients experiencing extended survival or tumor control. However, challenges remain, including relatively low immunogenicity, difficulty in identifying effective antigens, high costs of personalized approaches, and suboptimal delivery efficiency [76]. Addressing these limitations requires innovative adjuvants, advanced peptide modification technologies, and synergistic combination therapies to improve the clinical translational potential of peptide vaccines.

## 5. Cell-Based Vaccines

Tumor cell-based vaccines are prepared by ex vivo processing of tumor cells derived from either the patient (autologous) or other donors (allogeneic). Their primary goal is to present TSAs to the immune system, thereby activating antitumor responses aimed at preventing, controlling, or treating cancer. Common types of cell-based vaccines include whole tumor cell (WTC) vaccines, DC vaccines, and genetically engineered cellular vaccines. This section focuses primarily on WTC vaccines and DC vaccines.

Whole tumor cell vaccines are developed by processing autologous or allogeneic tumor cells using methods such as irradiation, heating, or freezing, which inactivate their oncogenic potential. The processed cells are then reintroduced into patients, either alone or in combination with adjuvants, to stimulate a robust immune response. The primary advantage of WTC vaccines over other vaccine types lies in their broad antigen spectrum. These vaccines present not only TAA, TSA, and tumor neoantigens, but also cryptic antigens. Cryptic antigens are those that are either not expressed or expressed at low levels in normal cells but are exposed in tumor cells due to genetic mutations, splicing abnormalities, or transcriptional changes. Such antigens include oncogenic fusion proteins, splice variants, endogenous retroviral elements, and post-translationally modified proteins. In tumor immunotherapy, cryptic antigen vaccines target these antigens, guiding the immune system to recognize and attack tumor cells, thus overcoming the limitations of traditional vaccines that rely solely on known antigens [77].

Preclinical studies have shown that WTC vaccines can effectively induce tumor immune responses. However, like most cancer vaccines, their clinical efficacy remains limited. Only a few clinical trials have shown positive results. For instance, in a prospective phase III study, the efficacy of an autologous tumor lysate-loaded DC vaccine in glioblastoma patients was evaluated. The results indicated that the vaccine, when combined with standard treatment, significantly extended patient survival compared to standard treatment alone, with statistical significance [78]. In another prospective randomized controlled trial, the safety and efficacy of Gemogenovatucel-T (Vigil, an autologous tumor cell vaccine) were examined in patients with newly diagnosed advanced ovarian cancer. The experimental group demonstrated a recurrence-free survival (RFS) of 11.5 months (95% CI: 7.5—not reached), compared to 8.4 months (7.9–15.5) in the placebo group (HR: 0.69, 90% CI: 0.44–1.07; one-sided *p* = 0.078), with no grade 3 or higher adverse events reported in the vaccine treatment group [79]. In another study, a DC/multiple myeloma (MM) fusion vaccine combined with Lenalidomide maintenance therapy was compared for its effects on immune response and clinical outcomes in MM patients after autologous hematopoietic stem cell transplantation (auto-HCT). While the vaccine did not significantly improve the complete remission (CR) rate at one year (52.9% vs. 50%, *p* = 0.37), it successfully induced the expansion of MM-reactive T cells, indicating tumor-specific immune effects. Furthermore, single-cell transcriptomic analysis revealed CD8 T cell clonal expansion and shared dominant TCR types, suggesting that the vaccine promoted improvements in the immune microenvironment [80]. While WTC vaccines show potential in cancer immunotherapy, they face several challenges, including insufficient immunogenicity, tumor heterogeneity, immune escape mechanisms, inadequate adjuvant use, and difficulties in production. To address these challenges, solutions include the use of immunogenic cell death inducers, the adoption of allogeneic WTC vaccines, the combination of multiple immune adjuvants, the incorporation of ICIs, and the standardization of production processes. These strategies can enhance both the immunological efficacy and production efficiency of WTC vaccines, ultimately improving therapeutic outcomes [81].

Dendritic cells play a pivotal role in mediating and regulating immune responses. As professional APCs, DCs serve as a bridge between innate and adaptive immunity and are essential for shaping immune memory and modulating immune activation. Consequently, DC vaccines have garnered significant attention in tumor immunotherapy [82]. Early studies have demonstrated that DC vaccines are safe in lung cancer treatment and capable of enhancing immune responses, with notable clinical benefits in some patients [83,84,85,86,87]. However, most of these studies were small-scale, single-arm trials lacking control groups. Additionally, the immunogenicity of the targeted antigens was often suboptimal, and the vaccines were primarily administered to patients with advanced disease, resulting in limited therapeutic efficacy. Subsequent research has aimed to optimize these limitations. For instance, a prospective phase I clinical trial evaluated a novel DC vaccine in patients with stage I–IIIA NSCLC following surgical resection using a dose-escalation design. The results showed good tolerability, with no dose-limiting toxicities observed in the high-dose group, and a significant improvement in patients’ quality of life [88]. Another study compared the safety and efficacy of autologous DC vaccines combined with cytokine-induced killer (CIK) cells versus chemotherapy alone in treatment-naïve patients with advanced NSCLC. Although the combination group experienced a higher incidence of mild adverse events such as rash and pruritus, it showed a significantly lower incidence of grade 3–4 fatigue (7.1% vs. 57.1%, *p* = 0.01) and a prolonged median progression-free survival (PFS) (6.9 vs. 5.2 months, *p* = 0.03) [89].

In a separate prospective, single-arm phase II trial, DC vaccination combined with chemotherapy was well tolerated in patients with advanced non-squamous NSCLC. Among the 44 patients in the intention-to-treat (ITT) population, the 2-year survival rate was 52.57%, with a median PFS of 8.0 months and an objective response rate (ORR) of 31.82%. Subgroup analysis showed that patients receiving high-dose DC vaccines had significantly better OS compared to those receiving low doses (*p* = 0.0038) [90]. Additionally, another study evaluated DC vaccines combined with chemotherapy in advanced NSCLC patients and reported an ORR of 34.8% (95% CI: 16.4–57.3%), a median PFS of 10.72 months (95% CI: 4.52–16.93), and a 1-year survival rate of 77.8% [91].

With the rapid advancement of bioinformatics, neoantigen-based vaccines have emerged as a major focus of research. Our team previously reported on the safety and efficacy of a neoantigen-loaded DC vaccine in heavily pretreated patients with advanced lung cancer (NCT02956551). A total of 12 patients were enrolled and received 85 doses, with a median of 4.5 doses per patient (range: 3–14). All treatment-related adverse events were grade 1–2, and no treatment delays due to toxicity occurred. The ORR was 25%, disease control rate (DCR) was 75%, median PFS was 5.5 months, and median OS was 7.9 months [92]. Building on this work, we further investigated the combination of neoantigen-loaded DC vaccines and ICIs in patients with ICI-resistant advanced lung cancer. Among the 15 patients screened, 9 received at least one dose of the neoantigen DC vaccine. One patient died due to pericardial effusion and another from COVID-19. Three patients progressed after completing five doses, while four continued treatment. As of March 2025, the median PFS was 10.5 months, and the median OS was 14.37 months. Among patients who received 10–12 doses, 44.4% (*n* = 4) experienced a >20% reduction in target lesion size, with two patients achieving reductions of over 50% (NCT06329908).

Despite the promising outcomes of tumor-derived DC vaccines—especially those incorporating personalized neoantigens—their clinical application remains limited by complex manufacturing processes, high costs, and lengthy production cycles compared to mRNA or peptide vaccines. Future studies are needed to address these challenges and improve the scalability and accessibility of personalized DC vaccine therapies.

## 6. Viral Vector Vaccines

Viral vector vaccines hold significant promise in cancer immunotherapy. These vaccines employ genetically engineered viruses to deliver tumor antigens into host cells, thereby inducing tumor-specific immune responses. Compared to other vaccine platforms, viral vectors offer several advantages: strong immunogenicity, high antigen expression efficiency, the ability to carry large exogenous gene sequences, ease of genetic modification, and favorable characteristics for storage, transport, and combination with other therapies. As such, viral vector platforms have attracted increasing attention in cancer vaccine development [93]. Common viral vectors include adenoviruses, poxviruses, and herpes simplex viruses.

Viral vector vaccines have already demonstrated clinical efficacy in several solid tumors, such as melanoma and glioblastoma [94,95]. A recent phase I clinical trial evaluated the safety and efficacy of the oncolytic herpes simplex virus VG161 in patients with advanced hepatocellular carcinoma who previously received standard second-line therapy. Among 37 evaluable patients, the disease control rate was 64.86%, the objective response rate was 18.92%, median progression-free survival (PFS) was 2.9 months, and median OS reached 12.4 months—surpassing historical control data and suggesting good translational potential for this viral vector vaccine [96].

Moreover, a study published in *Cell* in 2025 reported the development and clinical application of a novel genetically engineered oncolytic virus, NDV-GT. This vaccine was constructed using the Newcastle Disease Virus (NDV) platform and incorporated the porcine α1,3-galactosyltransferase (α1,3GT) gene into the viral genome, enabling tumor cells to express αGal antigens. This design mimics xenografts, triggering a hyperacute rejection response in the host and significantly enhancing antitumor immunity. Clinical data from 20 patients with advanced, treatment-refractory solid tumors who completed therapy showed a disease control rate of 90%, including one complete response, six partial responses, and eleven cases of stable disease, highlighting the vaccine’s promising clinical potential [97].

In the context of lung cancer, several early-phase clinical studies have evaluated the safety and efficacy of viral vector vaccines. For example, a phase I trial investigated an adenovirus vector delivering a MUC1/CD40L fusion protein in patients with advanced adenocarcinoma. Among the 21 patients enrolled, no dose-limiting toxicities were observed. The most common adverse events were injection site pain and rash, indicating a favorable safety profile [98]. Another phase II study evaluated the efficacy of the recombinant viral vector TG4010, administered either alone or in combination with chemotherapy in patients with stage III/IV NSCLC. Results showed a median OS of 12.7 months in the combination group and 14.9 months in the monotherapy group. TG4010 was well tolerated, with the most frequent adverse events being mild to moderate injection site reactions, flu-like symptoms, and fatigue [99]. Ongoing trials are currently investigating the combination of TG4010 with immune checkpoint inhibitors in advanced lung cancer (NCT02823990).

Viral vector vaccines demonstrate encouraging potential in the immunotherapy of lung cancer. Some candidates have shown favorable safety profiles and the ability to induce antitumor immune responses in early-phase clinical trials. However, challenges such as pre-existing anti-vector immunity, limited efficacy in certain patient populations, and insufficient large-scale clinical evidence still restrict their broader application. Future efforts should focus on optimizing viral platforms, developing combination strategies, and designing personalized antigen payloads to enhance their therapeutic value in lung cancer treatment.

## 7. Challenges and Potential Solutions for Lung Cancer Vaccines

Although cancer vaccines have made some progress in clinical applications, their overall efficacy still falls far behind that of infectious disease vaccines. This is mainly due to two reasons. First, cancer vaccines are mostly therapeutic, requiring the induction of immune responses in the compromised immune systems of cancer patients, whereas infectious disease vaccines are typically preventive and applied to healthy individuals, making it easier to induce immune responses. Second, factors such as the immunosuppressive tumor microenvironment, immune tolerance, tumor antigen heterogeneity and low immunogenicity, inefficient vaccine delivery, and complex immune evasion mechanisms all pose significant challenges to the efficacy of cancer vaccines. To overcome these difficulties, it is essential to optimize the selection of target antigens, combine with other therapeutic approaches, enhance adjuvants, and improve vaccine delivery platforms to improve the clinical efficacy of cancer vaccines [100].

Small cell lung cancer (SCLC) and NSCLC exhibit significant differences in immune microenvironment, tumor biology, and immune response to therapy, leading to distinct vaccine designs and treatment strategies. SCLC has a high mutational burden and immunogenicity, but due to its aggressive nature, immunosuppressive microenvironment, and complex immune evasion mechanisms, monotherapy with vaccines is often ineffective and typically requires combination with chemotherapy, immunotherapy, or other treatments. In contrast, NSCLC tends to grow more slowly, has a more complex immune microenvironment, and some patients respond well to immune checkpoint inhibitors. Therefore, vaccine designs for NSCLC focus more on targeting personalized neoantigens and enhancing cell-mediated immune responses.

### 7.1. Antigen Selection: A Fundamental Hurdle in Vaccine Development

The selection of appropriate target antigens is the primary and most critical step in vaccine development. An ideal tumor antigen should meet at least two essential criteria: (1) High specificity, enabling precise recognition by the immune system; (2) High immunogenicity, sufficient to elicit robust immune responses. However, the high degree of tumor heterogeneity in lung cancer significantly limits the identification and broad applicability of universal antigens. This heterogeneity exists both inter-patient (between individuals) and intra-tumor (within different regions of the same tumor), and it is considered one of the key contributors to immune evasion and therapeutic resistance [13].

In this context, personalized neoantigen vaccines have emerged as a promising strategy to address tumor heterogeneity. Neoantigens are derived from tumor-specific somatic mutations and are typically absent from normal tissues, giving them high specificity and immunogenic potential [34]. At present, the identification of neoantigens relies on an integrated bioinformatics pipeline, encompassing next-generation sequencing, HLA typing, and immunogenicity prediction—processes that have been extensively reviewed in previous literature [101,102]. While clinical trials have shown that neoantigen vaccines can induce antitumor responses [72], the overall survival benefits observed remain modest in most studies. Thus, optimizing neoantigen selection and identification strategies remains a major challenge in this field [103]. Another significant limitation is the long production cycle and high cost associated with neoantigen vaccines. Developing personalized vaccines is technically demanding and involves multiple complex steps, often requiring several months—far exceeding the therapeutic “window of opportunity” for many patients with advanced lung cancer. For example, Ott et al. reported in 2017 that the design and production of a personalized melanoma vaccine took approximately 12 weeks on average [26]. Sahin et al. later proposed an mRNA-based vaccine platform that could reduce this timeframe to about 6 weeks [27]. Nonetheless, the overall cost of developing personalized neoantigen vaccines can still reach tens of thousands of U.S. dollars, and currently, there are no viable solutions for large-scale cost reduction [57].

To address these limitations, shared neoantigens (also known as public neoantigens) have been proposed as a feasible alternative. These are antigenic peptides derived from recurrent driver mutations or common cancer-associated alterations that are present across multiple patients. Shared neoantigens can be efficiently presented by HLA molecules and are capable of eliciting tumor-specific T cell responses [104]. Compared with fully personalized vaccines, shared neoantigen vaccines offer several advantages.

#### 7.1.1. Faster Production and Lower Cost

Shared neoantigens can be pre-screened and validated, enabling the development of off-the-shelf vaccine products that significantly shorten production timelines and reduce cost.

#### 7.1.2. Broader Patient Coverage

For example, the EGFR L858R mutation is frequently observed in lung cancer patients carrying HLA-A*11:01 and can be targeted in a semi-personalized treatment model [105,106].

#### 7.1.3. Strong Antitumor Potential

As these antigens originate from driver gene mutations, they tend to be stable over time, reducing the risk of immune escape due to antigen loss [107].

Despite these advantages, the clinical application of shared neoantigen vaccines still faces several challenges. First, HLA diversity significantly limits the immunological coverage of any single neoantigen: the wide variation in HLA-A, -B, and -C alleles among individuals leads to differences in antigen presentation efficiency and immune responsiveness [108]. Second, tumor heterogeneity and dynamic antigen expression must be considered. Studies have shown that only about 20% of neoantigens are shared between primary and metastatic lung cancer lesions, and antigen loss or downregulation during tumor evolution may impair immune recognition [109]. Finally, the process of verifying immunogenicity is technically complex and costly. Even when mutation frequencies are high, determining whether peptides can be effectively presented by MHC molecules and activate T cells requires mass spectrometry and functional assays, both of which are time-consuming and resource-intensive [110].

To overcome these barriers, future research should adopt systematic strategies integrating immunopeptidomics, bioinformatics, and clinical profiling. On one hand, multi-antigen combination strategies could incorporate several high-frequency driver mutations to improve coverage across diverse HLA backgrounds. On the other hand, coupling high-throughput mass spectrometry with TCR-based functional validation platforms may significantly improve the accuracy and efficiency of neoantigen screening. Moreover, leveraging artificial intelligence-driven predictive models, based on population-level HLA distribution data and cancer mutational landscapes, could enable region- or ethnicity-specific vaccine design [111,112]. From a clinical perspective, exploring combination strategies—such as co-administration with ICIs—may further enhance the intensity and durability of T cell responses, offering new avenues for the broader application of shared neoantigen vaccines in lung cancer therapy.

### 7.2. Immunosuppressive Tumor Microenvironment Significantly Limits Vaccine Efficacy

The tumor microenvironment (TME) is a complex system comprising various immune components. Among them, immunosuppressive cells such as regulatory T cells (Tregs) and M2-polarized macrophages create a suppressive ecological niche that protects tumors from immune attack. These cells directly or indirectly inhibit the activity of effector T cells by releasing immunosuppressive signals, thereby limiting the immune system’s ability to recognize and eliminate tumor cells [112,113,114].

Combination therapy is widely considered one of the most effective strategies to overcome tumor resistance. Previous studies have shown that radiation therapy not only promotes T cell generation and trafficking to tumor sites—enabling recognition and elimination of tumor cells—but also modulates the immune status of the TME [115]. Moreover, radiotherapy has been found to promote the release of TAAs, effectively acting as an in situ *vaccine* [116]. It can also enhance the maturation of dendritic cells [117] and their antigen-presenting capacity [118] while upregulating multiple cytokines and chemokines, thereby further improving antitumor immune responses [119]. The combination of radiotherapy with ICIs has demonstrated significant promise in both preclinical and clinical studies, showing enhanced efficacy compared to monotherapy [120]. Conventional chemotherapy agents can also exert immunomodulatory effects by enhancing tumor immunogenicity and activating immune effector cells [121]. The combination of chemotherapy with ICIs has already been widely adopted in the treatment of various malignancies, including lung cancer [9,10,11,12].

As a novel immunotherapeutic modality, cancer vaccines are increasingly being integrated into multimodal treatment strategies involving radiotherapy, chemotherapy, and immunotherapy. In the START trial, although vaccine-based maintenance immunotherapy failed to meet its primary endpoint, subgroup analysis revealed that patients who received prior chemoradiotherapy experienced significantly improved outcomes in the vaccine group compared to the placebo group [122]. The TIME study evaluated the efficacy of TG4010—a modified vaccinia virus-based vaccine targeting MUC1 and IL-2—in combination with chemotherapy versus placebo in first-line treatment of advanced NSCLC. Results showed that the combination significantly improved PFS with a favorable safety profile [123]. Additionally, a phase II trial investigated the combination of vaccine therapy and bevacizumab following definitive chemoradiotherapy in patients with non-squamous NSCLC. The results demonstrated a median OS of 42.7 months in the combination group, with good overall treatment tolerance [124]. The combination of cancer vaccines with immune checkpoint inhibitors is also being explored in early-stage clinical studies across multiple tumor types, including hepatocellular carcinoma [125,126], melanoma [127,128], and lung cancer [74,129]. These studies support the feasibility and safety of vaccine-based combination therapies. However, variations in efficacy across different patient populations, along with the potential for immune-related adverse events, warrant further investigation. Additionally, the mechanisms underlying these synergistic effects remain to be fully elucidated, and the optimal combination strategies require validation through extensive clinical research.

Vaccine adjuvants are another key element in enhancing immune responses. Adjuvants not only play a critical role in initiating and modulating immune responses, but also significantly influence antigen delivery efficiency and the durability of immune protection. A recent systematic review summarized the mechanisms of action and delivery platforms for different types of adjuvants [130]. Nonetheless, the optimal pairing of vaccine platforms with specific adjuvants remains an active area of investigation.

### 7.3. Optimization of Vaccine Delivery Platforms

Optimizing vaccine delivery platforms is another key factor in ensuring vaccine efficacy. An effective delivery system can enhance the immune system’s recognition and response to antigens, preserve antigen stability, and minimize side effects. Among current technologies, LNPs represent the most clinically translatable delivery platform for cancer vaccines, particularly mRNA-based vaccines. LNPs offer advantages such as high encapsulation efficiency, biodegradability, and intracellular mRNA stability. Their safety and efficacy were well validated in COVID-19 vaccines and have since been rapidly adopted in the field of cancer immunotherapy [131,132]. Recent studies have demonstrated that co-delivery of adjuvants, optimization of administration routes, and fine-tuning of lipid composition can significantly enhance mRNA stability, improve cellular uptake, and boost immune activation [133]. In terms of targeted delivery, Selective Organ Targeting (SORT) technology enables precise mRNA delivery to specific organs such as the lungs, liver, or spleen, thereby improving CD8^+^ T cell-mediated immune responses [134,135]. Additionally, multi-layered “programmable” LNP structures are under investigation. These systems incorporate modular domains designed to enable spatiotemporal control of release and organ-specific localization [136].

However, despite their success in COVID-19 vaccines, LNP-based delivery systems still face multiple challenges in the context of cancer vaccines—particularly for lung cancer applications. These include: Intrinsic immunogenicity of LNPs, which may provoke unintended inflammatory responses; Reduced mRNA stability and transfection efficiency under the oxidative stress typical of the tumor microenvironment; Difficulties in maintaining batch consistency and purity during large-scale manufacturing; Limited targeting capability for lung and lymphoid tissues; And a lack of comprehensive evaluation of long-term safety and post-vaccination immune memory [137,138]. These issues currently hinder the broader clinical translation of LNP-based vaccines for solid tumors. In the future, advances in SORT-based targeting, multifunctional adjuvant carriers, high-throughput and scalable manufacturing techniques, and novel administration routes—such as inhalation or subcutaneous injection—are expected to drive the development of more efficient, safe, and precise delivery systems, ultimately laying a solid foundation for the clinical application of lung cancer vaccines.

## 8. Conclusions

As an emerging form of immunotherapy, lung cancer vaccines hold promise due to their ability to activate specific immune responses and induce durable immunological memory. When combined with ICIs, radiotherapy, chemotherapy, or immunologic adjuvants, these vaccines may enhance synergistic antitumor effects and potentially overcome the limitations of monotherapies, offering improved survival benefits for patients. However, the clinical translation of lung cancer vaccines still faces considerable challenges, including the immunosuppressive tumor microenvironment, immunological heterogeneity, and the need to improve vaccine immunogenicity and delivery efficiency. With continued advancements in bioinformatics, high-throughput sequencing, and delivery technologies, lung cancer vaccines are rapidly evolving toward more precise and personalized approaches. Future research should focus on efficient antigen screening; identification and validation of predictive biomarkers; optimization of combination therapy strategies; and deep exploration of tumor immune microenvironment regulation. By leveraging multidisciplinary collaboration across immunology, oncology, and bioengineering, the field can accelerate progress toward developing lung cancer vaccines that are more effective, safer, and clinically accessible.

## Figures and Tables

**Figure 1 vaccines-13-00836-f001:**
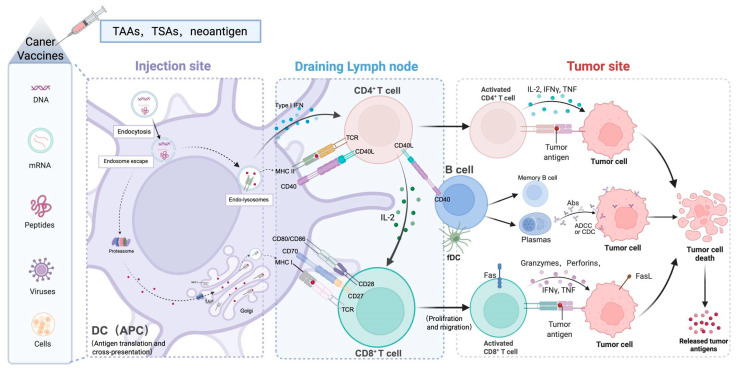
Immune responses induced by cancer vaccines. Upon vaccination, antigen-presenting cells (APCs), particularly dendritic cells (DCs), capture tumor antigens—derived from tumor-associated antigens (TAAs), tumor-specific antigens (TSAs), or neoantigens—and process them via cytosolic or endosomal pathways. These antigens may originate from DNA, mRNA, peptides, viral vectors, or tumor cell-based vaccines. After internalization by endocytosis, nucleic acids escape the endosome and are translated into proteins, which are degraded by proteasomes and presented on major histocompatibility complex (MHC) molecules. Mature DCs subsequently migrate to secondary lymphoid organs, such as draining lymph nodes, where they interact with naïve T cells. T cell activation involves three key signals: Signal 1 is delivered through recognition of the antigen–MHC complex by the T cell receptor (TCR); Signal 2 involves the binding of co-stimulatory molecules such as CD80/CD86 or CD70 to their respective ligands (e.g., CD28, CD27); and Signal 3 is mediated by cytokines and chemokines, including interleukin-2 (IL-2), which promote T cell proliferation, differentiation, and migration. Activated CD4^+^ helper T (Th) cells assist in the activation and differentiation of B cells via CD40–CD40L interactions and contribute to the production of cytokines such as IL-2, interferon-gamma (IFN-γ), and tumor necrosis factor (TNF). These cytokines enhance the cytotoxic function of other immune cells. Follicular dendritic cells (fDCs) further support B cell maturation, leading to the generation of antibody-secreting plasma cells and memory B cells. Effector CD8^+^ cytotoxic T lymphocytes (CTLs), primed by antigen-presenting DCs, infiltrate the tumor site and recognize tumor antigens presented on MHC I molecules of tumor cells. They mediate tumor cell killing through the release of perforin, granzymes, and Fas ligand (FasL), resulting in apoptosis and the release of additional tumor antigens, which may further amplify the immune response. Simultaneously, antibodies produced by plasma cells bind to tumor cell surfaces and mediate antibody-dependent cellular cytotoxicity (ADCC) and complement-dependent cytotoxicity (CDC), further contributing to tumor cell elimination.

**Table 1 vaccines-13-00836-t001:** Summary of Ongoing Clinical Studies on Lung Cancer Vaccines.

Grouping	NCT Number/	Population	Line	Interventions	Phases	Enrollment	Primary Outcomes	Secondary Outcomes	Target Antigen
Peptide Vaccine	NCT05950139	Advanced ALK + NSCLC	1	Vaccine	I/II	12	TRAEs/Immune Response	/	/
NCT03879694	Metastatic Neuroendocrine Tumors (NETs)	2	Vaccine + Octreotide Acetate + Sargramostim	I	14	TRAEs	ORR/PFS/DOR/Immune Response/Rate of Progression	Survivin
NCT01720836	NSCLC	1	Vaccine + PolyICLC	I/II	30	Immunologic Response	TRAEs/PFS/OS/Anti-MUC1 Immunity	Mucin1
NCT06751901	Advanced NSCLC	3	Vaccine + ICIs + Radiotherapy	II	10	ORR/DCR/TRAEs	PFS/OS	Personalized Neoantigen
NCT05254184	Advanced ALK + NSCLC	2	Vaccine + Nivolumab + Ipilimumab + PolyICLC	I	12	TRAEs	PFS/Immune Response	KRAS
NCT05269381	Advanced Solid Tumors	2	Cyclophosphamide + Neoantigen Peptide Vaccine + Pembrolizumab + Sargramostim	I/II	36	TRAEs	The percentage of Immunogenicity Responders and Trial Completers	Personalized Neoantigen
NCT06095934	EGFR + Advanced NSCLC	2	Chemotherapy + Vaccine + PD-1 Monoclonal Antibody	II	20	ORR	PFS/OS	Personalized Neoantigen
NCT06472245	Metastatic NSCLC	3	Arm A: OSE2101Arm B: Docetaxel	III	363	OS	/	P53, HER-2, CEA, MAGE-2 and MAGE-3, and One Pan-HLA DR Binding epitope
NCT06202066	Metastatic Neuroendocrine Tumors	2	Part 1: Temozolomide + VaccinePart 2: Arm a: TemozolomideArm b: Temozolomide + Vaccine	II	132	Part 1: PFS/TRAEsPart 2: PFS	Part 1: ORR/OS/TTPPart 2: TTP/TRAEs/ORR/OS	Survivin
NCT04266730	Advanced NSCLC or SCCHN	2	Pembrolizumab + Vaccine	I	6	TRAEs	ORR/OS/PFS	Personalized Neoantigen
NCT05344209	Advanced or Metastatic NSCLC	1	PD-1/PD-L1-Treatment ± UV1 Vaccination	II	138	PFS	TRAEs	hTERT
NCT04298606	Stage IB-IIIA Lung Cancer	Postoperative	Vaccine	I	60	≥3 Grade TRAEs/Biomarker Analysis	quality of life score	EGF
NCT05104515	Locally Advanced or Metastatic NSCLC, Ovarian Cancer, and Prostate Cancer	≥2	Vaccine	I	36	TRAEs	ORR	Survivin
NCT06015724	Refractory NSCLC/Pancreatic Ductal Adenocarcinoma	≥2	Daratumumab + Nivolumab + Vaccine	II	54	ORR	TRAEs/PFS/OS/DOR	KRAS
NCT06752044	Advanced NSCLC	2	Radiotherapy + Pd-1 + Vaccine	/	10	ORR/DCR/TRAEs	PFS/OS	Personalized Neoantigen
NCT06751901	Advanced NSCLC	>2	Radiotherapy + Pd-1 + Vaccine	II	10	ORR/DCR/TRAEs	PFS/OS	Personalized Neoantigen
mRNA Vaccine	NCT03908671	Advanced Esophageal Cancer or NSCLC	/	Vaccine	I	24	TRAEs	DCR/PFS/OS	Personalized Neoantigen
NCT05142189	NSCLC	/	Vaccine alone or Vaccine + Cemiplimab/Docetaxel/Carboplatin/Paclitaxel/BNT316/Anti-B7-H3 Antibody/Anti-HER3 Antibody	I	220	DLTs/TRAEs	ORR/DOR/PFS/OS	MAGE A3, CLDN6, KK-LC-1, PRAME, MAGE A4, MAGE C
NCT06928922	Advanced Lung Cancer	≥2	Vaccine or Vaccine + PD-1	I	22	DLTs/MTD/TRAEs	ORR/DOR/PFS/OS	/
NCT05557591	Advanced NSCLC	1	Arm A: CemiplimabArm B: Cemiplimab + vaccine	II	100	ORR	DOR/PFS/OS/TRAEs	MAGE A3, CLDN6, KK-LC-1, PRAME, MAGE A4, MAGE C
DNA Vaccine	NCT04397003	ES-SCLC	1	Durvalumab + Chemotherapy + Vaccine	II	20	TRAEs	ORR/DOR/PFS/OS	Personalized Neoantigen
NCT05242965	Advanced NSCLC	Maintenance Therapy	Arm I: Vaccine + SargramostimArm II: Sargramostim	II	40	CD8^+^ TIL/TRAEs	Immune Response/ORR/PFS/OS	CD105/Yb-1/SOX2/CDH3/MDM2
DC Vaccine	NCT05886439	Advanced NSCLC or ES-SCLC	2	Vaccine + Pembrolizumab or Durvalumab	I	40	DLT/TRAES	ORR/DOR/PFS/OS	Personalized Neoantigen
NCT04147078	Postoperative Locally Advanced Gastric Cancer, Hepatocellular Carcinoma, Lung Cancer and Colorectal Cancer	Postoperative	Vaccine	I	80	DFS	OS/TRAEs	Personalized Neoantigen
NCT06329908	Advanced Lung Cancer	≥2	Vaccine + ICIs	I	20	TRAEs	ORR/PFS	Personalized Neoantigen
NCT05195619	Advanced or Recurrent Metastatic NSCLC.	Stable Disease after Immunotherapy or Targeted Therapy	Vaccine + ICIs or Targeted Therapy	I	16	TRAEs	ORR/DOR/PFS/OS	Personalized Neoantigen
NCT06752057	Advanced NSCLC	2	Radiotherapy + Pd-1 + Vaccine	/	10	ORR/DCR/TRAEs	PFS/OS	Personalized Neoantigen

**Abbreviation:** TRAEs: treatment-related adverse events, ORR: overall response rate, PFS: progression-free survival, DOR: duration of response, OS: overall survival, TTP: time to progression, DCR: disease-control rate, DLTs: dose-limiting toxicities, MTD: maximum tolerated dose, KRA: kirsten rat sarcoma viral oncogene homolog, P53: tumor protein p53, HER-2: human epidermal growth factor receptor 2, HER-3: human epidermal growth factor receptor 3, CEA: carcinoembryonic antigen, MAGE: melanoma antigen gene 2, MAGE A3: melanoma antigen gene A3, MAGE A4: melanoma antigen gene A4, MAGE C: melanoma antigen gene C, hTERT: human telomerase reverse transcriptase, PRAME: preferentially expressed antigen in melanoma, CLDN6: claudin 6, KK-LC-1: kita-kyushu lung cancer antigen-1, Pd-1: programmed cell death protein 1, ICIs: immune checkpoint inhibitors, CD8^+^ TIL: cluster of differentiation 8 tumor-infiltrating lymphocytes, Poly-ICLC: polyinosinic-polycytidylic acid stabilized with polylysine and carboxymethylcellulose, mRNA: messenger ribonucleic acid, DNA: deoxyribonucleic acid, SCCHN: squamous cell carcinoma of head and neck, NSCLC: non-small cell lung cancer, ALK: anaplastic lymphoma kinase, EGFR: epidermal growth factor receptor.

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
