# Peer review of "Prospects and Challenges of Lung Cancer Vaccines"

_vaccines, 2025, doi:10.3390/vaccines13080836_

Round 1
Reviewer 1 Report
Comments and Suggestions for Authors
The Review study by Lin Z. et al.,” Prospects and Challenges of Lung Cancer Vaccines, vaccines-3740351”, is a standard compilation of data related to cancer vaccines with particular focus on lung cancer and personalized neoantigen vaccines. The authors describe the vaccine development steps from sequencing/antigen selection to vaccine delivery platforms and, importantly, their opinion on probability of success of these type vaccines is well balanced along with the discussion of particular features of several vaccine platforms. Perhaps, the inclusion of additional data on whole tumor cell-based vaccines and, adding the reference of the only neoantigen-based phase 3 clinical trial study (Weller M., et al., Lancet Oncol. 18, 1373-1385, 2017) might be useful. Also, the elimination of the word “great” from the sentence “…lung cancer vaccines hold great promise due to their ability to activate specific immune responses…” in Conclusion, could be more adequate.
Author Response
comment 1: Perhaps, the inclusion of additional data on whole tumor cell-based vaccines.
reply 1: Thank you for your suggestion; we have added the additional data on whole tumor cell-based vaccines in the manuscript, specifically on page 10, lines 360 to 403.
comment 2: adding the reference of the only neoantigen-based phase 3 clinical trial study (Weller M., et al., Lancet Oncol. 18, 1373-1385, 2017) might be useful.
reply 2: Thank you for your suggestion; we have added relevant references, including similar studies, as references 77-81 in the manuscript.
comment 3: Also, the elimination of the word “great” from the sentence “…lung cancer vaccines hold great promise due to their ability to activate specific immune responses…” in Conclusion, could be more adequate.
reply 3: Thank you for your suggestion; the word "great" has been deleted.
Reviewer 2 Report
Comments and Suggestions for Authors
This review article provides a thorough overview of the current landscape of lung cancer vaccines within the broader context of lung cancer treatment. It opens by underscoring the persistent global burden of lung cancer, which remains one of the most common and deadly malignancies despite standard interventions such as surgery, chemotherapy, and radiotherapy. These conventional therapies have yielded only modest improvements in survival rates, leaving prognosis poor and highlighting the urgent need for more effective treatment strategies.
The article then examines advances in immunotherapy, particularly the transformative impact of immune checkpoint inhibitors (ICIs) on cancer treatment. However, it notes that the clinical benefits of ICIs in lung cancer are limited by key challenges, including tumor heterogeneity, a highly immunosuppressive tumor microenvironment (TME), and the ability of tumors to evade immune detection. These factors collectively hinder both the efficacy and durability of immune-based therapies for many patients.
Against this backdrop, the review explores the potential of cancer vaccines as an innovative and complementary immunotherapeutic strategy. Unlike passive immunotherapies, cancer vaccines are designed to actively stimulate the host immune system to identify and eliminate tumor cells. Their high specificity, minimal toxicity, and capacity to induce long-lasting immune memory make them an appealing approach, especially given the significant issues of immune escape and resistance seen in lung cancer.
The main body of the article summarizes recent clinical advances in lung cancer vaccine development, covering both antigen-specific and personalized vaccine strategies. It also discusses major obstacles to their broader clinical adoption, such as suboptimal immunogenicity, difficulties in selecting suitable antigens, and the need for more effective delivery platforms. The review further explores strategies to address these limitations, including combination therapies with ICIs, the introduction of novel adjuvants, and personalization based on tumor neoantigen profiles.
In conclusion, the article positions cancer vaccines as a promising and rapidly evolving area in lung cancer research, with the potential to overcome some limitations of current therapies and improve long-term patient outcomes.
However, there seems to be a trend of incorporating cryptic antigens into immuno-oncology is no longer optional, the notion for advancing scalable, effective, and equitable cancer therapies. With enabling technologies like immunopeptidomics, ribosome profiling, and modular therapeutic platforms now available, the field may:
- Map cryptic antigens across cancer types, stages, and HLA backgrounds at scale, supported by functional assays to validate immunogenicity.
- Integrate cryptic targets into platforms such as mRNA vaccines and TCR-T therapies—especially in combination with canonical antigens—to enhance immune responses.
- Understand their origin, including the roles of alternative translation and ribosomal fidelity in generating these peptides.
- Clarify their immunobiology, including whether they are naturally recognized, how vaccination enhances visibility, and their susceptibility to immunoediting.
The review will improve if the authors could include some lines and explanation of cryptic antigens, here is a bit explanation based on recent reviews such as: https://lnkd.in/gawqKYyc. They do not need to use this particular mini-review and it is just an example.
However, there seems to be a trend of incorporating cryptic antigens into immuno-oncology is no longer optional, the notion for advancing scalable, effective, and equitable cancer therapies. With enabling technologies like immunopeptidomics, ribosome profiling, and modular therapeutic platforms now available, the field may:
- Map cryptic antigens across cancer types, stages, and HLA backgrounds at scale, supported by functional assays to validate immunogenicity.
- Integrate cryptic targets into platforms such as mRNA vaccines and TCR-T therapies—especially in combination with canonical antigens—to enhance immune responses.
- Understand their origin, including the roles of alternative translation and ribosomal fidelity in generating these peptides.
- Clarify their immunobiology, including whether they are naturally recognized, how vaccination enhances visibility, and their susceptibility to immunoediting.
Author Response
comment: The review will improve if the authors could include some lines and explanation of cryptic antigens.
reply: Thank you for your suggestion; we have added the additional data and relevant referrences on cryptic antigens in the manuscript, specifically on page 10, lines 366 to 373.
Reviewer 3 Report
Comments and Suggestions for Authors
This manuscript reviews the current status, challenges and opportunities for lung cancer vaccines.
The authors have thoroughly reviewed the literature and effectively subdivided the topic into logical sections. They have also covered both basic, preclinical and clinical data.
When examining this manuscript from the perspective of readers of “Vaccine”, many of whom are more experienced in infectious disease vaccines, the reader is left somewhat puzzled as to why cancer vaccines have thus far been far less successful than their traditional counterparts. This is mainly because therapeutic cancer vaccines are typically employed when the patient’s immune system is already immunosuppressed. This point was mentioned, but needs to be emphasized as a separate section and in more detail. How might this situation be overcome? Given the differences in genetics and biology, might these solutions vary between small and non-small cell lung cancer?
Author Response
comment 1: This point was mentioned, but needs to be emphasized as a separate section and in more detail. How might this situation be overcome?
reply 1: Thank you for your valuable feedback. We have made the requested revision and emphasized this point in more detail as a separate section. Please refer to page 13, lines 506 to 517 of the manuscript for the changes.
comment 2: Given the differences in genetics and biology, might these solutions vary between small and non-small cell lung cancer?
reply 2: Thank you for your insightful comment. While the approaches may differ between small cell lung cancer (SCLC) and non-small cell lung cancer (NSCLC), research on tumor vaccines for SCLC is currently limited, and further studies are needed to explore this. We have provided clarification on this in the manuscript, specifically on page 13, lines 518 to 527.